# From Laboratory to Everyday Life:
# Personalized Stress Prediction via Smartwatches

**Batuhan Koyuncu** [1]  **Aleyna Dilan Kıran** [1]  **Katja Heilmann** [2]  **Laith Hamid** [3]  **Anja Buder** [4]  **Veronika Engert** [2]
**Martin Walter** [4]  **Isabel Valera** [1]

## Abstract

Accurate prediction of stress in everyday life is essential to prevent chronic stress and maintain health and well-being through early and personalized intervention. With the goal of enabling reliable prediction suitable for everyday life, we present MUSTP, a two-stage machine learning pipeline designed to predict stress from low-resolution heart rate (HR) and high-resolution electrocardiography (ECG) measurements from commercial smartwatches. Our model is pretrained with labeled data collected in a controlled laboratory stress study. Subsequently, we *transfer* the model for everyday use, enabling it to operate with everyday smartwatch data in various environments. The model transfer strategy effectively addresses the domain shift from laboratory data to highly imbalanced smartwatch data and allows personalization. The empirical results on smartwatch data show that MUSTP can predict stress everyday with an F1 score of 52%, despite the measurements having sparse labels for stress.

## 1. Introduction & Related Work

Stress phenomena can be defined as the psychological responses of an individual to stressors, which can have significant implications for health and wellness when in a chronic state. Some established detrimental health outcomes associated with chronic stress include cardiovascular conditions, obesity, and mental health disorders such as depression and anxiety (Khan & Khan, 2017; McEwen, 2017; Ippoliti et al., 2013). These outcomes emphasize that there is an increasing necessity for reliable methods of stress prediction and monitoring in daily life (Al-Atawi et al., 2023; Can et al., 2019).

---

[1]Saarland University, Saarbrücken, Germany [2]Institute of Psychosocial Medicine and Psychotherapy, University Hospital Jena, Jena, Germany [3]Univesity of Jena, Jena, Germany [4]University Clinic for Psychiatry and Psychotherapy, University Hospital Jena. Correspondence to: Batuhan Koyuncu <koyuncu@cs.uni-saarland.de>.

*Accepted at the 1st Machine Learning for Life and Material Sciences Workshop at ICML 2024.* Copyright 2024 by the author(s).

Physical signals including electrocardiography (ECG), heart rate (HR), and heart rate variability (HRV) provide valuable information about how someone's body responds in the stress states compared to relaxed states (Kinnunen et al., 2020). Recent advances in machine learning and deep learning have been applied to stress prediction using ECG, HR, and HRV (Haque et al., 2023; Ramírez et al., 2023). Utilizing ECG presents a promising strategy for stress prediction, as it captures the heart's electrical signals and the activity of the autonomic nervous system (Castaldo et al., 2016; Cinaz et al., 2013; Acharya et al., 2006; Healey & Picard, 2005). Moreover, ECG-based stress prediction offers several advantages in terms of applicability in different environments. Due to its non-invasive nature, ease of use, and low cost, ECG signals can be collected in real-time in the laboratory via sensors or in everyday life via commercial smartwatches (Velmovitsky et al., 2022; Siirtola, 2019).

Despite the possibility of collecting high-quality ECG data in both laboratory and daily life settings, individuals may exhibit varying responses to stressors across different environments and circumstances (Can et al., 2020). In the laboratory, the Trier Social Stress Test (TSST) (Kirschbaum et al., 1993) is a widely used experimental procedure to induce acute psychological stress in individuals. This controlled procedure provides well-defined stress labels and accompanying physiological signals, making it invaluable for training machine learning models. However, the necessity for personalizing these models becomes evident, as individuals frequently have physiological signals with different characteristics in relaxed and stress states (Islam & Washington, 2023; Tazarv et al., 2021).

On the other hand, in everyday life, people encounter stress in different circumstances such as work deadlines, relationship conflicts, and unexpected life events. The intensity, duration, and labeling of these real-life stress responses can significantly vary, unlike the controlled environment in the laboratory. This variation contributes to what is known as domain shift in machine learning (Zhou et al., 2021), where the statistical distribution of data differs between domains, posing challenges for model generalization and performance across varied contexts. Therefore, it is crucial to recognize and minimize the domain shift problem to enhance the ap-

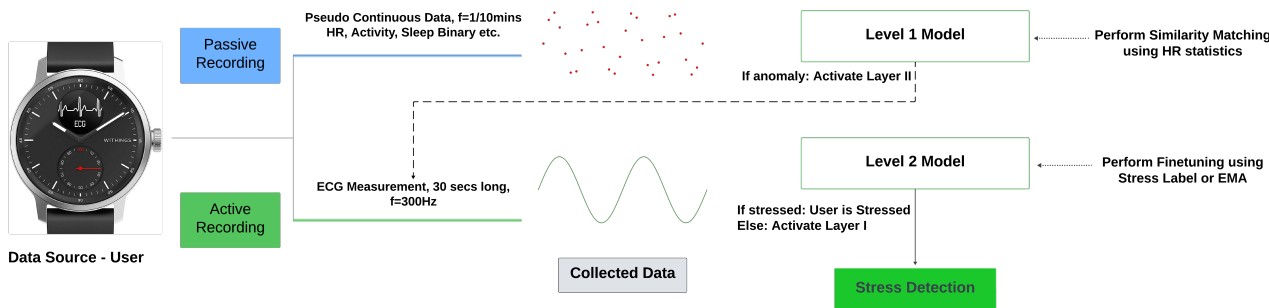

Figure 1: Sketch of the MUSTP framework.

plicability and accuracy of stress prediction models trained with laboratory data. This can be accomplished by fine-tuning the deep learning model for individual adaptation (Behinaein et al., 2021) or using transfer learning to deal with scarce data (Maxhuni et al., 2016).

In this paper, we propose a Multi-level Stress Predictor (MUSTP), a two-level ML pipeline that can operate with low-resolution HR and high-resolution ECG measurements for stress prediction. MUSTP minimizes user effort as it operates on low-resolution HR as the first level and only requires ECG measurements when needed. The proposed model leverages laboratory data collected with the TSST procedure for training. Afterward, the model is transferred to everyday life to operate with data collected from a commercial smartwatch (WITHINGS SCANWATCH) and subjective Ecological Momentary Assessment (EMA). We apply post-hoc optimization and finetuning to account for domain shifts and have personalized stress prediction. In the ablation study, we demonstrate the effects of model transfer on the model performance. Our empirical results on smartwatch data show that MUSTP achieves F1 score of 52% with a considerable improvement compared to the baseline model, even though only 31% of the measurements are labeled as stress.

## 2. Methodology

### 2.1. LABDATA- Dataset Description

The LABDATA dataset contains psychological data from 108 participants who underwent stress exposure via the TSST procedure. Due to recording issues, we eliminated 9 users; therefore, our dataset has effectively 99 users. The dataset includes ECG signals, salivary cortisol levels, and self-report measures of stress. ECG measurements are collected during the baseline and stress (TSST) conditions will be used as *hard labels of stress* for training our model. More details about the dataset can be found in Table 1, the source of HR modality described in Subsection 2.5. Each record of participants in the LABDATAis appropriately anonymized. The dataset is planned to be released in the future.

### 2.2. EVERYDAYDATA- Dataset Description

The EVERYDAYDATA dataset consists of 131 users whose data is collected for approximately two months via WITHINGS SCANWATCH. Users were advised to wear their smartwatches throughout the day and record an ECG along with completing an EMA three times per day during the designated windows of 7-10 AM, 12-3 PM, and 7-10 PM. Users are supposed to rest their arms on a table and hold the top electrode with their thumb and index finger for 30 seconds during ECG measurement. The measurements that are logged shortly after a workout or activity are discarded. For *(soft) stress labels*, we are using subjective stress from Affect Grid metrics from collected EMAs (see Appendix A.2 for more details), which are 19,289 in total. More details about the dataset can be found in Table 1. Similarly, each record of participants in EVERYDAYDATA is appropriately anonymized. The dataset is planned to be released in the future.

### 2.3. Preprocessing

In the following analysis, HR measurements are taken over 30-minute windows at a frequency of $1/600$ Hz, resulting in four measurements per window. ECG measurements, on the other hand, are taken in non-overlapping 30-second intervals, with the frequency specific to each dataset.

We preprocess the LABDATA dataset by applying a $0.5$ Hz high-pass Butterworth filter on collected ECG signals. Then, we perform R-peak detection to get the users' heart rates using the default algorithm in NEUROKIT2 (Makowski et al., 2021). For the EVERYDAYDATA dataset, we downsample ECG signals from 300 Hz to 250 Hz. After that, we apply a $0.5$ Hz high-pass Butterworth filter.

### 2.4. Proposed Model

We propose a two-level model, MUSTP as shown in Figure 1. In the following sections, we describe the model.

**Level 1** is an isolation forest-based model with the task of anomaly detection from HR measurements over 30 minutes with $1/600$ Hz. The baseline model (BASELINE-1)

| Dataset Name | #Users | Device Name | Stress Labels | Modality | #Baseline Measurements | #Stress Measurements |
|---|---|---|---|---|---|---|
| LABDATA | 99 | ZEPHYR BIOHARNESS 3.0 | TSST | ECG (250 Hz) | 3021 | 1980 |
| | | | | HR (1/600 Hz) | 3960 | 3960 |
| EVERYDAYDATA | 131 | WITHINGS SCANWATCH | EMA | ECG (300 Hz) | 7257 | 3396 |
| | | | | HR (1/600 Hz) | 6996 | 3263 |

Table 1: Overview of the datasets. We report the number of measurements for baseline and stress states.

is trained with synthetic HR measurements from baseline states in LABDATA. To transfer BASELINE-1 to the everyday environment, we create a pool of user-specific anomaly detectors using LABDATA. Then, when we apply Level 1 for a user's everyday settings, we select the model in the pool with baseline HRs that are statistically most similar to those of the current user from EVERYDAYDATA. We call this approach similarity matching (SM) (see Subsection 2.6).

**Level 2** is a Convolutional Long Short-Term Memory (LSTM) network-based binary classifier that classifies 30-second ECG signals into stress and non-stress. The baseline model (BASELINE-2) is trained with LABDATAfor the whole population. We transfer BASELINE-2 to everyday life setting by finetuning (FT) (see Subsection 2.7) it for each user in EVERYDAYDATA.

**MUSTP** Our model runs on Level 1 as a default mode. The hierarchical structure requires the first Level 1 Model to have a positive anomaly prediction to activate the second level. The combined hierarchical model has the following advantages:

- Operating with low-resolution HR signals in default mode, e.g. suitable to run with incoming data from commercial smartwatches.

- Higher reliability in the final prediction as input signal becomes high-resolution ECG measurement.

- Requiring minimum active participation of user as it is only for 30-second intervals in Level 2.

Hence, our model can be easily used in everyday life with commercial smartwatches following training in a laboratory setting. See Appendix A.3 for more details of the model.

### 2.5. Synthetic Data Generation

Originally, there was a lack of TSST-labeled HR data that is 30 minutes long with a frequency of $1/600$ Hz. To train the BASELINE-1 model, we synthesize a dataset comprising 30-minute-long heart rate signals. As a first step, the mean and standard deviation of HR measurements for each user were computed in both baseline and stress states. Using these statistics, we employ the NEUROKIT2 ECG simulation tool to generate 30-second-long synthetic ECG signals.

Subsequently, we calculate HR from the generated ECG signals. Through independent iterations of ECG generation and HR extraction, our objective is to produce realistic data with lower resolution. For each user in the legacy dataset, we have created 40 HR measurements for both baseline and stress states to model anomalies.

### 2.6. Similarity Matching

We use similarity matching (SM) for choosing the most suitable anomaly detector for a test user in the inference, rather than relying on a single anomaly detector trained on the entire LABDATAdataset.

To utilize SM, we first train an anomaly detector for each user training user $j \in \{1, \ldots, J\}$. The SM algorithm computes a distance between the test user ($i$) and each user in the training pool using statistics of heart rate measurements from baseline states. For simplicity, we assume baseline heart rate measurements of a user follow a univariate normal distribution with parameters $\mu$ and $\sigma$; therefore, we compute the similarity by using Bhattacharyya distance

$$d_{ij} = \frac{1}{4} \frac{(\mu_i - \mu_j)^2}{\sigma_i^2 + \sigma_j^2} + \frac{1}{2} \ln \left( \frac{\sigma_i^2 + \sigma_j^2}{2\sigma_i \sigma_j} \right), \quad (1)$$

where $i$ and $j$ denote indices of test and training users, respectively. After we compute distances $d_i = [d_{i1}, \ldots, d_{iJ}]$, we find the index $j^*$ that minimizes $d_{ij}$ as $j^* = \arg\min_{j \in \{1, \ldots, J\}} d_{ij}$. We then select the anomaly detector of the user $j^*$, with a corresponding threshold value of $\text{TH}_j^*$ for a target recall value.

### 2.7. Personalization of the Model

Finetuning (FT) a model can significantly enhance its performance for a specific domain or user. We follow a common approach to achieve this by fine-tuning only the last linear layer of the BASELINE-2 model. We rely on our deep model to extract informative features from the input signal. Therefore, instead of aligning the distribution of the input signal, we focus on tuning the model's final classification layer (Behinaein et al., 2021; Islam & Washington, 2023). This step updates the parameters of the linear layer considering the individual characteristics of input data. Furthermore, fine-tuning helps mitigate domain shift, as the baseline model is trained with LABDATA to extract robust, generalizable features, while the last layer can be finetuned

with EVERYDAYDATA for adjusting model outputs to the specific distribution of the everyday life domain. Further details on FT can be found in Appendix A.4.

### 2.8. Post-hoc Optimization of Thresholds

In order to use MUSTP as a full pipeline, it is important to select the best decision thresholds to increase model performance. To do so, we perform *post-hoc optimization* (PO) of user thresholds by maximizing the expected F1 score on a held-out dataset. This step is applied for Level 1 and 2 models separately. Thresholds after the optimization step are denoted as $\text{TH}_i^\dagger$ and $\text{TH}_i^\ddagger$ for the first and second level of the model for user $i$.

## 3. Experiments

### 3.1. Data and Training

We split our LABDATA data into $[70, 10, 19]$ users for training, validation, and testing. For Level 1, the amount of windows for HR measurements is $[2800, 400, 760]$ and $[2800, 400, 760]$, for baseline and stress states respectively, with each window has four HR measurements 10 minutes apart. For Level 2, there are $[2140, 308, 573]$ baseline ECG measurements and $[1400, 200, 300]$ stress ECG measurements, with each measurement being 30-second long. The windows are not overlapping in either modality.

Similarly, for EVERYDAYDATA, we have 131 users with the total of $[6996, 3263]$ windows of HR measurements for Level 1, and $[7257, 3396]$ ECG measurements for Level 2 in baseline and stress states respectively. All of the users in EVERYDAYDATA can be thought as test users.

BASELINE-1 and BASELINE-2 denote baseline models trained and validated with corresponding users of LABDATA. In order to check effect of the finetuning and post-hoc optimization of the thresholds, we use $40\%$ of the data collected from the test user over time in the respective dataset. For a comparable performance comparison, we report the metrics on the remaining $60\%$ of the data.

### 3.2. Results

In this section, we report the performance of BASELINE-1, BASELINE-2 models for the test users of LABDATA and EVERYDAYDATA to investigate effects of model transfer steps. We present receiver operating characteristic (ROC) and precision-recall (PR) curves. For comparison purposes, the area under the curve (AUC) and average precision (AP) are reported with the corresponding curves. The performance metrics are computed after aggregating the individual results of binary classification, i.e. summing TP, FP, TN, FN values from each user. Lastly, we provide an ablation study, in which we test the full pipeline of MUSTP with different combinations of transfer steps.

**Results with Level 1 in Laboratory Environment** For our BASELINE-1 model, we present ROC and PR curves in Figure 2 as baseline. For evaluating model transfer, we apply SM for BASELINE-1 and report the same metrics for the transferred model. Our approach of SM increases AUC for both ROC and PR curves.

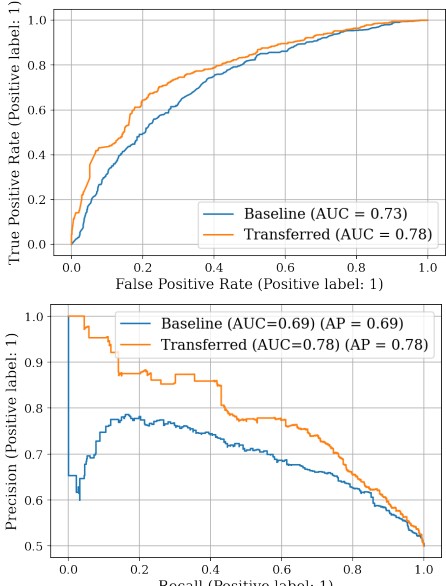

Figure 2: LABDATA, ROC and PR curves for Level 1

**Results with Level 2 in Laboratory Environment** For our BASELINE-2 we present ROC and PR curves in Figure 3. For evaluation of model transfer, we apply FT for BASELINE-2. Our model for Level 2 has $77\%$ accuracy for the test users with a threshold of $0.3$, which is decided on the validation set. After FT, the accuracy increases to $82\%$.

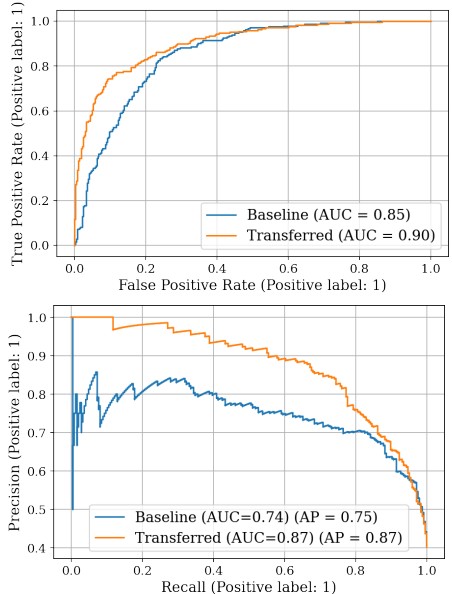

Figure 3: LABDATA, ROC and PR curves for Level 2

**Results with Level 1 in Everyday Environment** We present our results of BASELINE-1 model with EVERYDAY-DATA. For model transfer, we apply SM for BASELINE-1 without any further training. Our results are shown in Figure 4 with minimum improvement observed in ROC and PR.

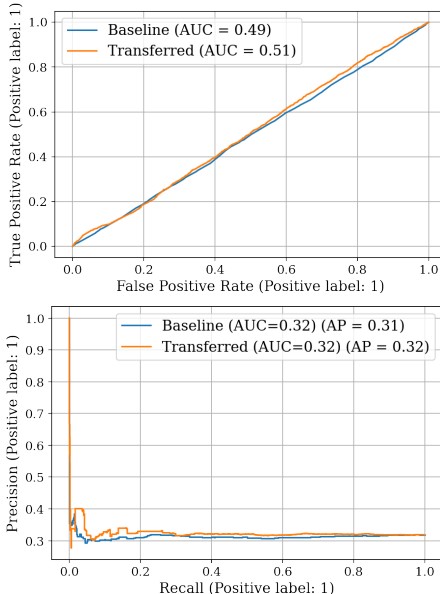

Figure 4: EVERYDAYDATA, ROC and PR curves for Level 1

**Results with Level 2 in Everyday Environment** In this part, we present our results of BASELINE-2 model with EVERYDAYDATA. For model transfer, we apply FT on BASELINE-2 by using $40\%$ percent of user data. Our evaluation on $60\%$ test data shows that model transfer increases AUC of PR and ROC curves as shown in Figure 5.

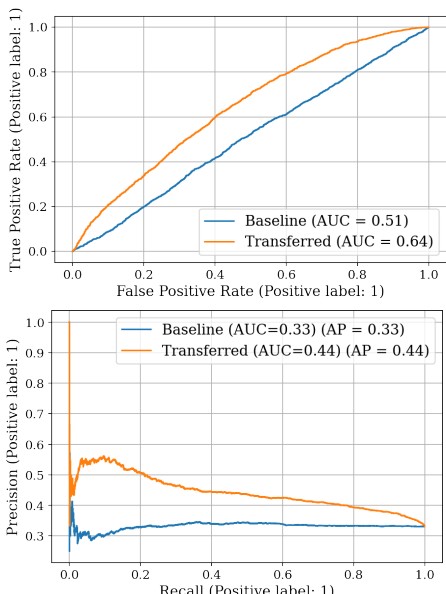

Figure 5: EVERYDAYDATA, ROC and PR curves for Level 2

**Results for MUSTP in Everday Environment** In this part, we present our results of MUSTP model with EVERYDAY-DATA. To evaluate the full model pipeline's performance, we need to use both HR and ECG measurements along with an EMA answer. When we consider the evaluation set corresponding to $60\%$ of EVERYDAYDATA dataset, it is observed that $31\%$ of data is labeled as stress. For further improvement, we perform PO for selecting user-specific thresholds that maximize F1 score after FTstep, using initial $40\%$ of user data in EVERYDAYDATA dataset for both steps. Thresholds after PO are denoted as $\mathrm{TH}_i^\dagger$ and $\mathrm{TH}_i^\ddagger$ for the Level 1 and 2 models for user $i$. We report a summary of models in Table 2.

| **Model Name** | **Models** | **Thresholds** |
|---|---|---|
| MUSTP Baseline | BASELINE-1 | $0.472$ |
| | BASELINE-2 | $0.3$ |
| MUSTP Transferred | BASELINE-1 + SM | $\mathrm{TH}_j^*$ |
| | BASELINE-2 + FT | $0.3$ |
| MUSTP Transferred + Opt | BASELINE-1 + SM +PO | $\mathrm{TH}_i^\dagger$ |
| | BASELINE-2 + FT + PO | $\mathrm{TH}_i^\ddagger$ |

Table 2: Overview of the models and their thresholds.

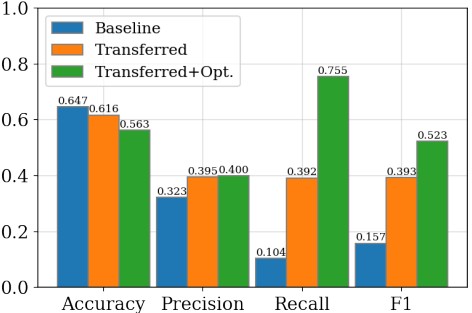

Figure 6: Comparison of Models in Everyday Environment

In Figure 6, we show the aggregated performance of MUSTP with baseline, transferred (SM+ FT), and transferred and optimized (SM+ FT+ PO) versions. Our results show that model transfer with optimization has the greatest F1 score among all models. The aggregation process involves summing the binary stress prediction outputs for each window across all users. We also show the distribution of the performance metrics among users in Figure 7.

## 4. Conclusion

In this work, we proposed MUSTP, a two-level ML pipeline for predicting stress in everyday environments using commercial smartwatches. Our model minimizes user involvement during inference mode in everyday life in order to increase its applicability and ease of use in real life. Our

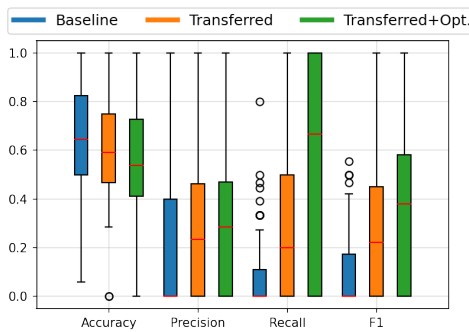

Figure 7: Comparison of Models in Everyday Environment

results show that MUSTP after model transfer has 52% F1 score with a considerable improvement on the baseline.

For future work, we plan to extend our current scheme using online reinforcement learning-based approaches to learn user-specific adaptive decision thresholds aiming for increased personalization without requiring further training or offline optimization. Lastly, we want to improve the reliability of our EMA-based soft labels using both Affect Grid and questionnaires and decide the contribution of each component by investigating the correlation between cortisol levels and soft labels.

## 5. Acknowledgments

The authors acknowledge the support by the German Federal Ministry of Education and Research through the Cello project under the project number: 16SV8590.

We thank Dr.-Ing. Mario Aehnelt and Nicola Marlene Drüeke from Visual Assistance Technologies, Fraunhofer-Institut für Graphische Datenverarbeitung IGD Rostock, for their assistance in developing the models and processing EMAs, Dimitri Kraft for his initial model developments, and Paul Burggraf and Hannes Schenk from Thryve, mHealth Pioneers GmbH, Berlin, for application development, data processing and storage. We also appreciate the contributions of Prof. Dr. Birgit Derntl, Sabrina Eutebach, and Juliane Köstlin from the Department of Psychiatry and Psychotherapy, University of Tübingen, for data collection in the Cello field study.

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

# A. Appendices

## A.1. Laboratory Experimental setup

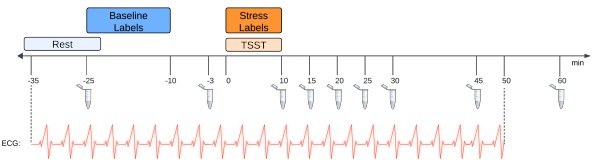

Figure 8: TSST Time Protocol with the corresponding label samples for each window for later classification

The TSST procedure involved an approx. 30-minute preparation period, followed by a 5-minute speech task and a 5-minute mental arithmetic task in front of a panel of judges. After the stressor, the recovery period starts, which takes approx. 38 minutes. Upon arrival at the laboratory, subjective stress questionnaires and cortisol samples were collected at -25, -3, 10, 15, 320, 45, and 60 minutes of the procedure. The ECG recording was conducted between -35 and 50 minutes. The experiment is illustrated in Figure 8.

## A.2. Soft Labels from EMAs

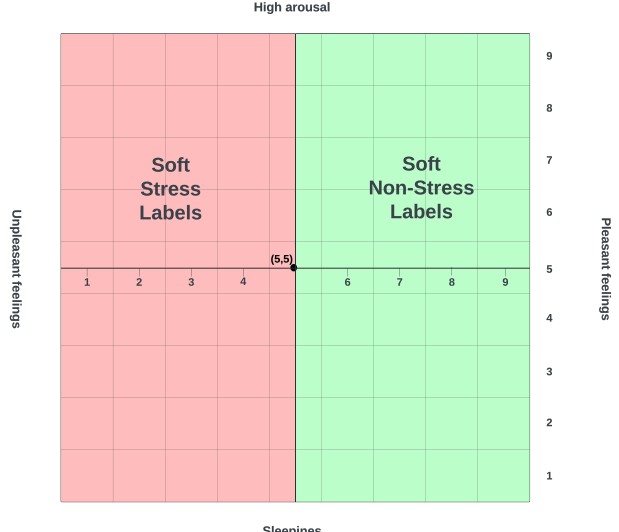

Figure 9: Defined stress regions in EMA grid

The subjective stress assessment of EMA is carried out by using Affect Grid (EMA GRID) (Russell et al., 1989). EMA GRID has an empty 9x9 grid with dimensions of pleasant-unpleasant and arousal-sleepiness. We soft label the left side of the grid as stress (unpleasant feelings) as shown in Figure 9. The EMA answers are collected through the Pulsatio Application.

## A.3. Model Architectures and Training

We describe our model for Level 1 in Table 3. We fit the isolation forest-based BASELINE-1 model using synthetic baseline data (baseline HR measurements) from LABDATA dataset. In order to compute the contamination parameter, we use HR measurements from baseline and stress states.

Table 3: Description of Isolation Forest Model

| Isolation Forest | Description |
|---|---|
| Number of Estimators | 200 |
| Contamination | 1e−5 |
| Max Samples | auto |

We describe our model for Level 2 in Table 4. We train BASELINE-2 model using LABDATA. We train BASELINE-2 with a learning rate of 1e−3, total epoch of 250, batch size of 32 and early stopping with 25 epochs.

Table 4: Description of Convolutional LSTM Network Architecture

| Layer | Description |
|---|---|
| conv | Input channel: 1, Output channel: 50, Kernel size: 150 |
| ReLU Activation | Applied after each conv layer |
| Pooling | Max pooling, Pooling size: 200 |
| Regularization | Dropout, Dropout rate 0.5 |
| Batch Normalization | Applied after dropout |
| LSTM Layer 1 | Input size 36, Output size 32 |
| LSTM Layer 2 | Input size 32, Output size 16 |
| Flattening | Flatten |
| Fully Connected Layer | Input size 800, Output size 1 |
| Output Activation | Sigmoid |

## A.4. Details of Personalization with Model Finetuning

For the Level 2 model, we perform finetuning on the last linear layer of BASELINE-2 model for 10 epochs with a learning rate of 1e−4 with a batch size of 16 for each test user using 40% percent of their data.

