# OpenReview forum: "From Laboratory to Everyday Life: Personalized Stress Prediction via Smartwatches"
_ICML.cc/2024/Workshop/ML4LMS — ML4LMS Poster_

### Official Review · Reviewer_yuot · 2024-06-11
**Interesting paper on transfer learning for stress prediction from heart rate and ECG data**

**Rating:** 6
**Confidence:** 3

**Review:**

The authors propose a 2-level machine learning pipeline for stress prediction from heart rate and ECG data. The first level is based on anomaly detection using low-frequency HR measurements. The second level comprises a convolutional LSTM network processing ECG signals from a smartwatch. The authors collect two large datasets 1) in a lab-controlled environment (99 participants); and 2) in a real-world scenario (131 participants) and leverage a form of transfer learning to transfer models trained on lab data to the real-world case.

The paper is potentially interesting and the results seem promising. However, there are a few issues with the paper, mainly regarding clarity on the methodology (i.e. signal pre-processing and the baseline methods the authors compare their results against).

I would kindly ask the workshop organizers to check if the submission falls under the areas of interest of the workshop, as it does not involve biology or chemistry, but the paper seems to be rather oriented around biomedical engineering.

Main comments

- Regarding level-1 model, please clarify whether 1 HR measurement is taken every 10 minutes (1/600 Hz) and if so, the length of activity that is recorded in order to estimate HR.

- Please clarify exactly what the user needs to do in order to accomplish high-resolution ECG measurement, once an anomaly has been detected in level-1 model.

- Section 2.7: "This step updates the parameters of the linear layer considering the individual characteristics of input data. Furthermore, fine-tuning helps mitigate domain shift, as the baseline model is trained with LABDATA to extract robust, generalizable features, while the last layer can be finetuned with EVERYDAYDATA for adjusting model outputs to the specific distribution of the everyday life domain.": according to the reviewer's experience, domain adaptation methods for biomedical signal processing are typically applied on the raw signal, which is first transformed and then passed into the network (e.g. the Euclidean alignment method used in this space for inter-subject EEG classification: https://doi.org/10.1109/TBME.2019.2913914). Could you please clarify why you chose to adopt a head fine-tuning approach and why you chose to fine-tune only the last layer of the network?

- Regarding the results presented in Figure 6: if 40% of the dataset has been used for fine-tuning, and another 40% has been used for hyper-parameter tuning (i.e. threshold selection), do results shown in this figure correspond to only 20% of the data? Are model comparisons shown in this figure done on the same test data? Furthermore, if this is just 20% of the data, can you please explicitly mention in the text the number of positive and negative samples in the test sets?

- Related to the above, please clarify if test scores are aggregated 1) within participants (e.g. across processing windows) and 2) across participants. I would encourage the authors to show error bars in this figure, or even better boxplots (which are non-parametric). If both types of aggregation has been performed, I believe it makes sense to compute average scores across processing windows within participants and show in the figure the variation of average scores across participants.

- Please clarify for every model / case / sensor modality what is the length of the processing window and what is the window increment in case processing windows are overlapping, otherwise clarify. I find it slightly hard to follow whether the train/val/test data samples presented in the text (e.g. Section 3.1) correspond to number of processing windows or sampled data points.

- Classification accuracy is mentioned as a metric in the text (e.g. Section 3.2  & Figure 6), but given that the dataset is highly-imbalanced, this is a very poor metric, so I would encourage authors to remove it altogether.

- The baseline methods that the authors compare their method against requires significant clarification. For example, what is a baseline Baseline-1 model in Figure 2? Is the baseline model trained on the same participant or not? How is the transferred model different to the baseline one (in terms of training)? Same question applies to Figure 4.

- Related to the above, what is the difference between Baseline and Transferred Baseline-2 model in Figure 3? Please clarify which users each of the two models has been trained on. Same question applies to Figure 5.